# Current Status of Epitranscriptomic Marks Affecting lncRNA Structures and Functions

**DOI:** 10.3390/ncrna8020023

**Published:** 2022-03-28

**Authors:** Henry E. Miller, Mirolyuba Ilieva, Alexander J. R. Bishop, Shizuka Uchida

**Affiliations:** 1Department of Cell Systems and Anatomy, UT Health San Antonio, San Antonio, TX 78229, USA; millerh1@livemail.uthscsa.edu (H.E.M.); bishopa@uthscsa.edu (A.J.R.B.); 2Greehey Children’s Cancer Research Institute, UT Health San Antonio, San Antonio, TX 78229, USA; 3Bioinformatics Research Network, Atlanta, GA 30317, USA; 4Center for RNA Medicine, Department of Clinical Medicine, Aalborg University, DK-2450 Copenhagen SV, Denmark; mirolyubasi@dcm.aau.dk; 5May’s Cancer Center, UT Health San Antonio, San Antonio, TX 78229, USA

**Keywords:** epitranscriptomics, gene expression, lncRNA, RNA-seq

## Abstract

Long non-coding RNAs (lncRNAs) belong to a class of non-protein-coding RNAs with their lengths longer than 200 nucleotides. Most of the mammalian genome is transcribed as RNA, yet only a small percent of the transcribed RNA corresponds to exons of protein-coding genes. Thus, the number of lncRNAs is predicted to be several times higher than that of protein-coding genes. Because of sheer number of lncRNAs, it is often difficult to elucidate the functions of all lncRNAs, especially those arising from their relationship to their binding partners, such as DNA, RNA, and proteins. Due to their binding to other macromolecules, it has become evident that the structures of lncRNAs influence their functions. In this regard, the recent development of epitranscriptomics (the field of study to investigate RNA modifications) has become important to further elucidate the structures and functions of lncRNAs. In this review, the current status of lncRNA structures and functions influenced by epitranscriptomic marks is discussed.

## 1. Introduction

By definition, long non-coding RNAs (lncRNAs) are any ncRNAs that are longer than 200 nucleotides (nt). With the advancement of high-throughput techniques [microarrays, next generation sequencing (NGS), especially RNA sequencing (RNA-seq)], many lncRNAs have been discovered [1]. To date, a number of functions of lncRNAs have been proposed and experimentally validated; ranging from decoy, epigenetic, transcriptional, post-transcriptional, and translational controls [2,3,4,5]. The general understanding in the field is that lncRNAs exert their actions by binding to other macromolecules, which are DNA, RNA, and proteins [6,7]. Thus, it is essential to identify the potential binding partners to elucidate the mechanism of action of lncRNAs. To this end, the most popular method is using an affinity tag on an in vitro purified RNA and using this RNA as a bait to pull-down proteins/nucleic acids from cellular extracts. There are other more elaborated methods currently available, including ChIRP (Chromatin isolation by RNA purification), CHART (Capture Hybridization Analysis of RNA Targets), CLIP (cross-linking and immunoprecipitation), and RAP (RNA antisense purification), which are comprehensively reviewed elsewhere [8,9,10].

Just as DNA and proteins, RNA can be modified by a variety of enzymes. The classic example is the RNA modifications of ribosomal RNAs (rRNAs) and transfer RNAs (tRNAs), which affect the efficiency of translation [11,12,13]. To date, there are more than 170 RNA modifications known across species [14], which has opened up a new field of study called, epitranscriptomics [15,16], whose name is based on the well-studied field of DNA modification, epigenetics. Much of the concepts of epigenetics are applied to dissect the ever-growing field of epitranscriptomics, including the epitranscriptomic enzymes being categorized as writers, readers, and erasers. Among epitranscriptomic marks, the most well studied one in recent years is N^6^-methyladenosine (m^6^A), which is a methylation of nitrogen-6 position of adenosine (A) found in messenger RNAs (mRNAs) and non-protein-coding RNAs (ncRNAs). Other epitranscriptomics marks in mammals include the A-to-I RNA editing, 2′-O-methylation (2′-O-Me), N^1^-methyladenosine (m^1^A), 3-methylcytidine (m^3^C), 5-methylcytosine (m^5^C), N^7^-methylguanosine (m^7^G), pseudouridylation (Ψ) to name a few [17,18]. These epitranscriptomic marks affect all realms of RNA lifecycle, including splicing, subcellular localization, microRNA (miRNA) biogenesis and bindings, RNA stability, and translation efficiency [19,20]. More importantly, dysregulation of epitranscriptomic marks affect many diseases, including cardiovascular [21], liver [22], and neurodegenerative diseases [18] as well as cancers [23]. These epitranscriptomic marks are also found in lncRNAs [24]. Given that epitranscriptomic marks affect the binding between lncRNA and other macromolecules is still a matter of ongoing investigation, which this review aims to summarize.

## 2. Epitranscriptomic Marks Affect RNA Structures as in the Case of Immune Responses

RNA exists in a single-stranded (ssRNA) or double-stranded RNA (dsRNA) state. The balance between these states may be influenced by cellular conditions, such as stress and viral infection [25,26,27,28]. Furthermore, more than half of the human genome consists of repetitive sequences, such as those derived from transposons and ALU elements [29]. These repetitive sequences form palindromic repeats, resulting in the formation of dsRNAs [30]. To detect dsRNAs, there are several high-throughput methods available, including PARS (Parallel Analysis of RNA Structure) by sequencing RNA digested with RNases S1 and V1 that specifically recognize single-stranded RNA (ssRNAs) and dsRNAs, respectively [31]. Other methods to analyze RNA structures are DMS-Seq to label RNA structures by dimethyl sulfate (DMS) [32], LIGR-seq (LIGation of interacting RNA followed by high-throughput sequencing) to globally map RNA–RNA duplexes crosslinked in vivo [33], PARIS to detect dsRNA [34], RIC-seq (RNA in situ conformation sequencing) to globally profile intra- and intermolecular RNA–RNA interactions [35,36], SHAPE-Seq (selective 2′-hydroxyl acylation analyzed by primer extension sequencing) [37], and SHAPE-MaP (selective 2′-hydroxyl acylation analyzed by primer extension and mutational profiling) to chemically probe RNA by adding RNA-specific small molecules in cell culture [38,39]. Recently, a comprehensive RNA structure probing database, RASP, was released, which contains 18 species (e.g., animals, plants, bacteria, fungi, and viruses) and 18 different experimental methods measuring RNA secondary structures in a transcriptome-wide manner [40]. Furthermore, there are databases for epitranscriptomic marks (comprehensively reviewed in [41]), including RMBase v2.0 [42] and RMVar [43] that contain several epitranscriptomic marks for different organisms. It will be of great interest to further analyze the collected data sets by merging them with high-throughput data that map known epitranscriptomic marks. This will enable the analysis of the preferential distribution of each epitranscriptomic mark to ssRNAs and dsRNAs in different species (thus, evolutional-conservation, if any).

Upon viral infection, the innate immune system is triggered, which recognizes pathogen-associated molecular patterns (PAMPs, which are unique molecular ligands on or within microbes, including viral DNA and RNA) leading to activation of intracellular signaling pathways to initiate antiviral response [44,45]. These PAMPs are detected by the host through pattern recognition receptors, such as Nod-like receptors (NLRs), RIG-I-like receptors (RLRs), and Toll-like receptors (TLRs) [46]. In the case of RLRs, RIG-I senses short dsRNAs, while the RLR, MDA5 (melanoma differentiation-associated protein 5), detects long dsRNAs. These recognitions of PAMPs by RLRs are followed by MAVS (mitochondrial antiviral-signaling protein)-mediated activation of signaling cascades, including type I interferon responses [47,48,49]. The epitranscriptomic mark, m^6^A, plays active roles in innate immunity by reducing type I interferon production [50,51]. Winkler et al. reported that m^6^A marks deposited by the m^6^A METTL3 and read by the m^6^A reader YTHDF2 negatively regulate interferon response by facilitating the fast turnover of interferon mRNAs leading to viral propagation [50] (Figure 1A). Interestingly, increasing evidence suggests that lncRNAs are shown to be involved in virus infections and antiviral immune responses [52]. Furthermore, many lncRNAs have m^6^A marks [53,54,55], influencing secondary structures of lncRNAs. For example, *MALAT1* (metastasis associated lung adenocarcinoma transcript 1) is involved in inflammatory responses and innate immunity [56,57,58] along with its enzymatic processing product, *MALAT1*-associated small cytoplasmic RNA (*mascRNA*) [59,60,61,62,63]. These findings highlight that further investigation of epitranscriptomic marks on lncRNAs and their secondary structural changes may reveal the active involvement of lncRNAs in innate immunity. In this regard, it will be of high interest to understand the relationship between lncRNAs and another epitranscriptomic mark, pseudouridylation (Ψ) [64], as it is demonstrated recently in COVID-19 mRNA vaccines using N^1^-methylpseudouridine (m^1^Ψ) to increase their effectiveness [65].

A-to-I RNA editing is a type of epitranscriptomic mark that involves the RNA editing enzymes, ADARs [adenosine deaminases acting on RNA, consisting of three genes: *ADAR1*, *ADARB1* (*ADAR2*), and catalytically inactive *ADARB2* (*ADAR3*)], recognize dsRNAs to catalyze adenosine to inosine (A-to-I) conversion, mostly at ALU repeats and introns [21]. ALU repeats are ~300 bp that belong to the family of repetitive elements in primates. There are more than one million ALU repeats in primate genomes [66]. Two transcribed ALU repeats form a quasi-palindrome, which becomes double-stranded RNA to recruit ADARs to catalyze A-to-I RNA editing [67]. I is recognized as guanosine (G) by splicing and translational machineries as well as in reverse transcription reactions; allowing detection of A-to-G changes in RNA-seq reads when these reads are mapped to the reference genome [68]. Mutations in the human *ADAR1* gene result in the autoimmune disease, Aicardi-Goutières syndrome, while the whole-body knockout mice of *Adar1* results in embryonic death due to massive apoptosis and aberrant interferon induction, which can be rescued to live birth by ablating the RLRs, Mavs or Mda5 (melanoma differentiation-associated protein 5) [69,70,71]. Both ADAR1 and ADAR2 are important in differentiating self- from non-self dsRNAs [70,72,73] (Figure 1B). Furthermore, silencing of *ADAR1* in the human hepatocellular carcinoma cell line, HepG2, resulted in shifting of dsRNAs to ssRNAs at the transcriptome-wide level [74]. As many lncRNAs have A-to-I RNA editing sites [75,76], further characterization of RNA editing sites will uncover the secondary structures of lncRNAs, especially the conversion of A to I at the nitrogen-6 position of adenosine, which can be methylated as m^6^A, if not edited [77]. Thus, both epitranscriptomic marks, A-to-I RNA editing and m^6^A, could competitively affect the secondary structures of lncRNAs, thereby, influencing the binding of other macromolecules.

## 3. Understanding the Actions of RNA-Binding Proteins Is Important for Functions of lncRNAs

RNA-binding proteins (RBPs) bind ssRNAs or dsRNAs to modulate their stability and translation [78]. RBPs also bind lncRNAs as in the case of the lncRNA *NORAD* sequestering the RBP, PUMILIO, to regulate genomic stability [79,80,81]. Classically, RBPs are defined to have specific motifs called, RNA-binding domains (RBDs). Through proteome-wide screenings (e.g., RNA interactome capture), over 2000 RBPs have been identified, of which many lack known RBDs [82,83,84]. In regards to lncRNAs, the catalytic subunit of polycomb repressor complex 2, EZH2, is a good example of a protein binding to RNA, including lncRNAs, without known RBDs, although promiscuously [85,86]. Indeed, there are databases available to examine the binding of RBPs to lncRNAs, including CLIPdb [87], POSTAR2 [88], and starBase [89]. Thus, increasing studies now investigate the possible binding of lncRNAs to RBPs thereby regulating RNA metabolism and functions (possibly as RBP sponges to sequester the available RBPs in the cell), instead of RBPs merely functioning in the biogenesis of lncRNAs [90].

RBPs can either stabilize or degrade the bound RNA [91] (Figure 2). Given that mRNAs have different epitranscriptomic marks, whether such marks signal the recruitment of RBPs to the mRNAs that ultimately affect the mRNA stability needs further investigation. A recent systematic analysis of RBP-bound regions and A-to-I RNA editing sites suggest that such RNA-edited sites are preferentially bound by the RNA-stabilizing RBP, HuR (encoded by *ELAVL1* gene) [92]. This study further confirms the possible functional role of RNA-edited sites of individual genes in regards to mRNA stability via the action of HuR [93,94]. Furthermore, several lncRNAs are found to bind HuR, including the lncRNA, *LINC02381*, that stabilizes the 3′-untranslated region (UTR) of the *HOXC10* mRNA via HuR [95]. Interestingly, although there is no A-to-I RNA editing (based on the DARNED database [96,97], https://darned.ucc.ie, accessed on 2 February 2022), nor m^6^A site (based on the m6A-Atlas database [98], http://180.208.58.19/m6A-Atlas/, accessed on 2 February 2022) in the 3′-UTR of *HOXC10* gene, an 5-methylcytosine (m^5^C) site (based on the m5C-Atlas database [99], http://180.208.58.19/m5c-atlas/, accessed on 2 February 2022) is present in the 3′-UTR of *HOXC10* gene, which calls for further investigation of this region to understand the causal relationship between lncRNA, RBP, and epitranscriptomic marks.

Given that many epitranscriptomic readers are RBPs (e.g., m^6^A readers: YTHDF1-3, YTHDC1, YTHDC2, HNRNPC, RBMX (HNRNPG), IGF2BP1-3 [100,101]), it is not surprising that the RBP binding sites overlap that of epitranscriptomic marks on mRNAs to regulate mRNA stability. In the case of RBP binding to the m^6^A sites, a convenient tool called, m6Adecom, was introduced recently [102]. However, it is of utmost importance to experimentally validate the relationships among epitranscriptomic marks, RBPs, and lncRNAs by performing gain/loss-of-function experiments for each of the components to understand the causal relation between them, rather than simply performing and analyzing high-throughput data.

## 4. Factors Influencing lncRNA Mediated R-Loop Formation—Sequence, Structure and Chemical Marks

R-loops (RNA loops) are three-stranded nucleic acid structures formed from the hybridization of RNA and DNA, leading to a displaced single-stranded DNA (ssDNA). These structures typically result from nascent transcription at regions of high GC-skew, such as CpG islands in gene promoters [103]. When improperly regulated, R-loops can be pathological [104,105,106]. A well-described consequence of pathological R-loops is collision of the transcriptional machinery with the replisome (transcription-replication conflict) [107]. Not surprisingly, because of these deleterious consequences, most interest has focused on investigating factors involved in R-loop processing or dissolution, such as the RNase H1 and RNase H2 enzymes that specifically degrade the RNA in R-loops [108]. However, under basal conditions, R-loops are benign; and, in some cases, they play important physiological roles [109,110]. Recent studies also indicate that lncRNAs are involved in the formation of R loops, such as lncRNAs *APOLO* [111], *HOTTIP* [112], and *TERRA* [113,114,115,116,117,118,119], and enhancer RNAs (a subclass of lncRNAs arising from the enhancer regions of protein-coding genes) [120] (Figure 3).

A critical aspect of R-loop biology that may impact functionality and dynamics are characteristics of the RNA moiety in these structures. There is emerging evidence that m^6^A methylation [121,122,123] plays a key role in regulating R-loop dynamics. In a recent study, Yang et al. showed that the m^6^A writer, METTL3, is required for R-loop formation at m^6^A+ gene termination sites to prevent transcript read-through [123]. These results were extended in a subsequent study by Abakir et al., which found that m^6^A readers like YTHDF2 regulate R-loop levels to prevent genome instability in dividing cells [122]. Curiously, Kang et al. found in their recent work that a pleiotropic transcriptional regulator (either an activator or a suppressor depending on individual genes), tonicity-responsive enhancer binding protein (TonEBP), recognizes and binds R-loops, recruits METTL3, leading to RNase H1 recruitment to facilitate R-loop resolution [121]. In addition to m^6^A methylation, recent evidence indicates that RNA secondary structure [124,125,126] influences R-loop dynamics. de Almeida et al. demonstrated that the DDX1 RNA helicase resolves RNA G4 quadruplexes to promote R-loop formation and promote IgH class-switching recombination [126]. Moreover, Chakraborty et al. recently demonstrated that the DHX9 helicase unwinds RNA secondary structure, leading to R-loop formation [125]. Finally, similar to mutations in *ADAR1*, mutations in the R-loop-processing complex RNase H2 also results in Aicardi-Goutières syndrome [127], indicating a convergence of mechanism that may relate to both A-to-I editing and R-loop processing.

R-loops form readily both from protein-coding and non-coding RNA species, particularly lncRNAs [109,128]. However, unlike protein-coding RNA species, lncRNAs form R-loops both in cis (re-annealing to their DNA template) and in trans (annealing to a distal region). Antisense lncRNAs form cis R-loops that modulate the transcription of sense protein-coding genes [129,130,131,132,133]. For example, the lncRNA *VIM-AS1* (VIM antisense RNA 1) forms an R-loop, leading to chromatin remodeling and expression of the sense vimentin (VIM) gene [130]. In a similar example, the lncRNA *TARID* (TCF21 antisense RNA inducing promoter demethylation, also known as *EYA4-AS1*) forms an R-loop which leads to GADD45A recruitment, demethylation, and expression of *TCF21* mRNA [131]. Finally, the lncRNA *GATA3-AS1* (GATA3 antisense RNA 1) forms an R-loop which promotes permissive chromatin marks and expression of *GATA3* mRNA [129]. Conversely, lncRNA-mediated R-loops can also repress antisense transcription. The lncRNA *ANRASSF1* (RASSF1 antisense RNA 1) forms an R-loop that recruits PRC2 to repress the *RASSF1* gene [132]. Additionally, the lncRNA *COOLAIR* forms a cis R-loop to repress transcription at the *FLC* locus in Arabidopsis [133]. These examples illustrate how lncRNA forms cis R-loops to influence chromatin state and modulate transcription of protein-coding genes. In a different mechanism, Tan-Wong et al. demonstrated that sense transcription of protein-coding genes leads to cis R-loops that can promote antisense lncRNA transcription [134]. A proposed explanation for the action of cis R-loops in promoting antisense transcription is that the displaced ssDNA acts as a de novo promoter for general transcription factors [134]. However, the mechanism of trans-acting lncRNA R-loops is less clear.

lncRNA forms an R-loop in trans via the invasion of duplex DNA distal to the site of lncRNA transcription. Wahba et al. first described this phenomenon in a 2013 study, which found that lncRNAs form trans R-loops via the action of Rad51p in yeast [135]. Subsequent work in human cell lines showed that the lncRNA *TERRA* (telomeric-repeat-containing RNA) forms R-loops in trans at telomeres, also in a RAD51 (DNA repair protein)-dependent manner [136]. However, lncRNA R-loops may also act independently of the RAD51 trans mechanism. In budding yeast, the lncRNA *GAL* forms R-loops which promote rapid adaptation to changing nutrient availability [137]. Curiously, these lncRNA R-loops form in both cis and trans upon depletion of the RNA helicase Dbp2 [137], suggesting an alternative mechanism that does not depend on Rad51p. Moreover, in the Arabidopsis genome, the *APOLO* lncRNA forms an R-loop which regulates the PID gene in cis and multiple auxin-responsive target genes in trans to promote lateral root development [111].

Recent work suggests lncRNA formation of trans R-loops may also play a role in supporting long-range chromatin interactions and enhancer formation via the cohesin complex. The cohesin complex is a multiunit complex comprising the SMC1/3, RAD21, and STAG1/2 proteins. It forms a ring structure responsible for establishing and maintaining 3D chromatin architecture [138]. Previous work has shown that the cohesin complex subunits STAG1/2 bind to R-loops and co-localize with them genome wide [139], suggesting the possibility that R-loop/cohesin interactions may regulate 3D chromatin conformation at enhancers. In 2018, Tsai et al. found that an enhancer lncRNA (eRNA) forms a trans R-loop that recruits cohesin to regulate the activity of the *Myogenin* locus [140]. Moreover, genomic analysis suggests that eRNA R-loops may act in trans to facilitate enhancer-promoter interaction via Alu repeats [141]. Finally, a recent study showed that the lncRNA *HOTTIP* (HOXA distal transcript antisense RNA) forms trans R-loops and promotes cohesin binding and long-range enhancer interactions [112]. Taken together, these findings suggest a possible role for some lncRNA R-loops as trans-acting promoters of long-range chromatin looping via their interactions with the cohesin complex. However, future work is necessary to better understand this potential mechanism and reveal other potential roles for lncRNA R-loops.

## 5. Conclusions

While functional characterizations of lncRNAs have intensified in the past decade, the identification and characterization of various epitranscriptomic marks are still at their infancy as only m^6^A marks are highly studied. As it is evident that lncRNAs are modified by various epitranscriptomic enzymes, it is still difficult to understand what these epitranscriptomic marks mean for the functions of lncRNAs with largely unknown functions. Furthermore, a recent study about epitranscriptomic marks on tRNAs show that epitranscriptomic marks affect tertiary structures of tRNAs [142], suggesting that not only secondary structures but also tertiary structures of lncRNAs must be carefully analyzed for the presence of epitranscriptomic marks on lncRNAs. Given that sequence information alone cannot infer the functions of lncRNAs, careful inspections of secondary and tertiary structures of lncRNAs are necessary to uncover the functions of lncRNAs as their mechanisms of actions depend on their bindings to macromolecules. Further development of experimental techniques as well as re-analysis of published data from the perspective of epitranscriptomic marks and lncRNAs are necessary to uncover the causal relationships of marked lncRNAs to their binding partners, especially RBPs, and R-loop formation (Figure 4).

## Figures and Tables

**Figure 1 ncrna-08-00023-f001:**
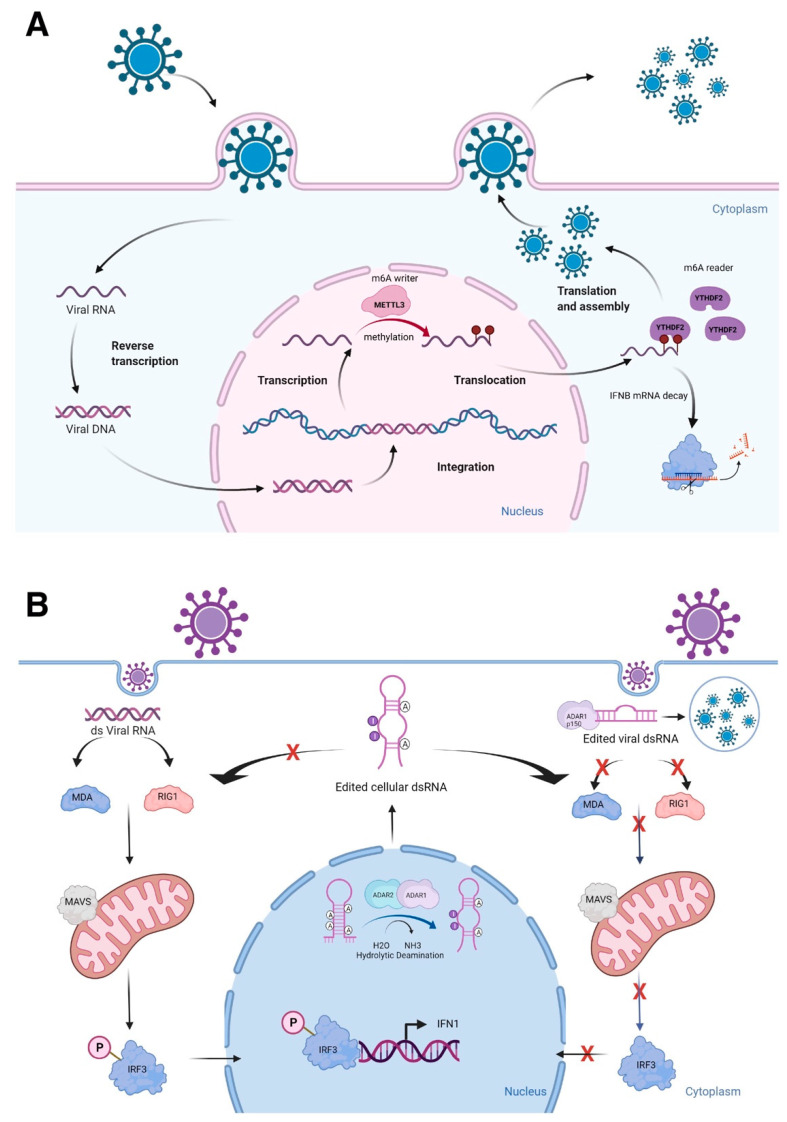
Epitranscriptomic marks and RNA structures in immune responses. (**A**) Viral RNA methylation deposited by the m^6^A writer METTL3 and read by the m^6^A reader YTHDF2 negatively regulate cellular defense response by facilitating the fast turnover of interferon mRNA leading to viral replication. (**B**) The role of ADARs in differentiating self-from non-self dsRNA by modulating canonical antiviral pathways induced by dsRNA. During an infection, the viral dsRNA enters into the cytoplasm. Non-edited dsRNA binds to MDA5 (melanoma differentiation-associated protein 5) and RIG-I (retinoic acid-inducible gene I like receptor). This complex activates MAVS (mitochondrial antiviral-signaling protein), leading to the phosphorylation of IRF3 (interferon regulatory transcription factor 3) and its translocation into the nucleus, thus inducing a type 1 interferon response. Endogenous cellular dsRNA that is generated during transcription is A-to-I edited by ADARs. The ADAR1 isoform p150 is cytoplasmic and is induced by interferon. It edits dsRNA either of viral or cellular origin. This dsRNA contains inosine and inhibits the activation of MDA5 and RIG-1, thus turning off the interferon response and apoptosis to prevent autoimmune reaction. However, this mechanism could favor virus replication, if it is not tightly regulated. Figure is created with BioRender.com, accessed on 15 March 2022.

**Figure 2 ncrna-08-00023-f002:**
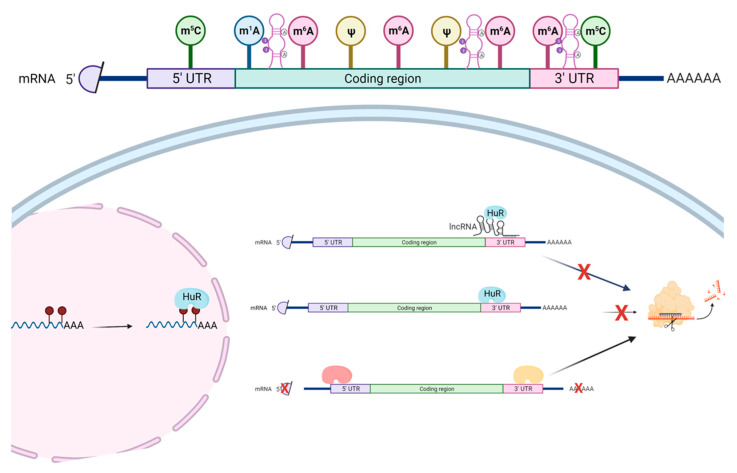
The roles of RBPs in the epitranscriptomic context. The most common epitranscriptomic marks include 5-methylcytosine (m^5^C), N1-methyladenosine (m^1^A), N6-methyladenosine (m^6^A), pseudouridine (Ψ), as well as A-I RNA editing. These marks affect the structures of RNA and, thus, influence the binding of RBPs. RNA-edited sites are preferentially bound by HuR—a RNA-stabilizing RBP. Furthermore, several lncRNAs, including *LINC02381*, are found to be bound by HuR, which stabilizes the 3′-UTR of the mRNA. Other RBPs could promote RNA degradation. Figure is created with BioRender.com, accessed on 16 March 2022.

**Figure 3 ncrna-08-00023-f003:**
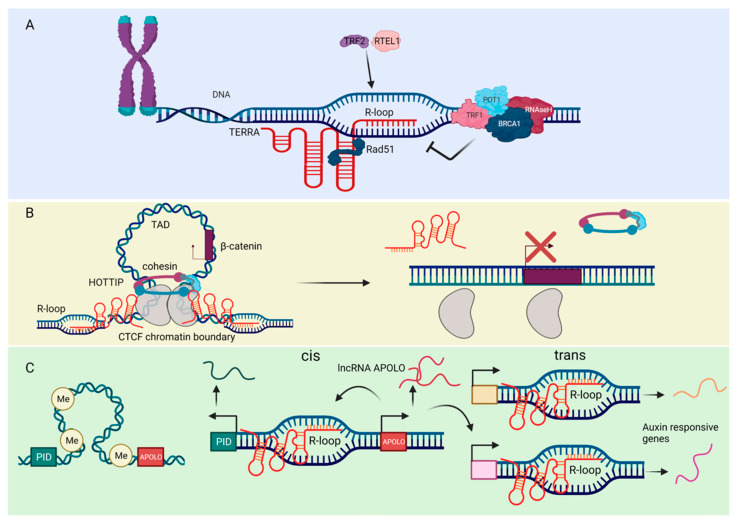
The roles of ncRNAs in the formation of R-loops. (**A**) The lncRNA *TERRA* binds to telomeres through direct base-pairing with telomeric DNA, forming the R-loop structure. The RAD51 DNA recombinase binds *TERRA* and thus catalyzes TERRA R-loop formation. The helicase RTEL1 and the telomeric shelterin component TRF2 also promote *TERRA* association with chromosome ends and stimulate R-loop formation. TRF1 and POT1 play a role in preventing the accumulation of such structures. The DNA recombination factor BRCA1 modulate *TERRA* binding to telomeres, preventing R-loop-associated telomeric DNA damage. *TERRA* R-loops are regulated by endonucleolytic cleavage activity of RNase H, which removes the RNA-DNA hybrids. (**B**) The lncRNA *HOTTIP* is highly expressed in acute myeloid leukemia. *HOTTIP* directly interacts with and regulates CTCF/cohesin complex and form R-loops. *HOTTIP*-mediated R-loops facilitate formation of topologically associating domain (TAD) to drive gene transcription of β-catenin. Eliminating *HOTTIP*-guided R-loops by targeting RNase H or removing CTCF-binding sites affect TAD structure, leading to alleviating leukemia severity. (**C**) The cis and trans regulatory mechanisms of the lncRNA *APOLO*. Under normal physiological conditions, *APOLO* is not transcribed due to hypermethylation of its promoter and chromosomal looping. However, during the lateral root development, the plant hormone Auxin induces the transcription of *APOLO*. The transcribed *APOLO* forms R-loops and thus modulates the chromatin looping in cis and trans, thereby enhancing transcription of auxin responsive genes and lateral root development. The figure was created with BioRender.com, accessed on 17 March 2022.

**Figure 4 ncrna-08-00023-f004:**
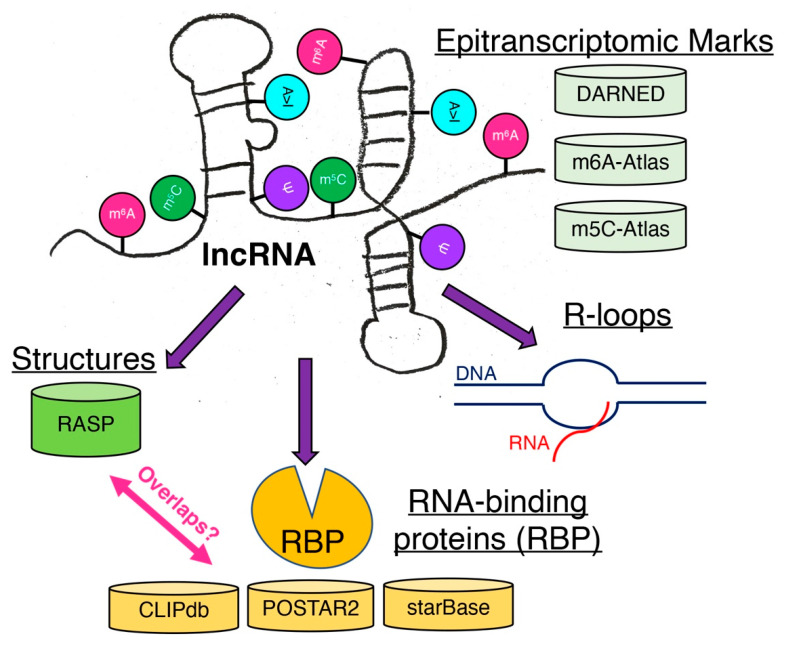
Possible modifications of lncRNAs. These epitranscriptomic marks affect the structures of lncRNAs as well as their binding to RNA-binding proteins (RBPs) and the formation of R loops. Some known databases are listed to facilitate further bioinformatic analysis to connect the information.

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
