# Peer review of "Current Status of Epitranscriptomic Marks Affecting lncRNA Structures and Functions"

_ncrna, 2022, doi:10.3390/ncrna8020023_

Round 1

Reviewer 1 Report

In recent years, a great process has been made in the research of RNA post transcriptional modifications, and hundreds of RNA modifications have been characterized. However, the role of RNA modifications in diverse cellular and biological processes are complicated. Here, Henry M et al. discussed the current understanding of the coupling and interdependence between lncRNA functions and their RNA modifications. The authors highlighted how various RNA modifications will influence lncRNA structures, lncRNA-protein binding and R-loops. While the authors have made an amount of great work, there are some points that could be further improved.

1. PARS was pretty old technology to study RNA secondary structures. It is better to update the newest methods in “Single- and double-stranded RNA play divergent biological roles” section (line 73), such as: RIC-seq, PARIS-seq, LIGR-seq.

2. Recently, some references also showed that RNA modifications will affect RNA tertiary structures. Could author also discuss the impact of RNA modifications on RNA tertiary structures. eg: RNA modifications stabilize the tertiary structure of tRNAfMetby locally increasing conformational dynamics. Nucleic Acids Res. 2022 Feb 28; 50(4): 2334–2349.

3. A-to-I editing mainly converts adenosine to inosine in double-stranded RNA regions. In Figure1, “A>I” should be labeled in dsRNA regions, not in single strand RNA regions.

4. In the R-loops section, the author should discuss more how RNA modifications influence lncRNA R-loops formation and their functions, not just present the role of lncRNA in R-loops. For example: how m6A regulates the R-loop stability.

N6-methyladenosine regulates the stability of RNA:DNA hybrids in human cells. Nat Genet. 2020 Jan;52(1):48-55.

m6A promotes R-loop formation to facilitate transcription termination. Cell Res. 2019 Dec;29(12):1035-1038.

5. Could the author prepare another Table or Figure to help readers to understand the RNA modifications and their potential biological functions more conveniently.

Reviewer 2 Report

This is a nice and timely review on an important topic, RNA modifications, also called epitranscriptomic marks, and their potential binding partners. It gives an overview of the field but without entereing in deep details. I only have a few suggestions on this manuscript.

The tittle should be changed to highlight the fact that most of the review is dedicated to RNA marks

I would like to see a more deep discussion on the effect of the A-I conversion on the process of exonization of Alu (and other) elements.

It would be also interesting that the authors gave a hint of the techniques used for detecting and charactyerizing the binding partners.  

Reviewer 3 Report

General Comment:

In it’s current format, the review is hard to read and the idea presented is inaccessible. The review seems to focus on too many concepts and loses coherence in the narrative. If my understanding stands correct, the general concept that this review describes is that non-coding RNA chemical modifications can alter its secondary structure, protein binding and its capacity to form R-loops. This suggests that epitranscriptomic modifications play a crucial role in regulating the life of a ncRNA. With recent progress in methods to identify/discover chemical modifications in RNAs in a high-throughput manner (Nat Rev Genet. 2017 May;18(5):275-291.), it is becoming increasingly important to understand the functional consequence of these epitranscriptomic marks. In this light, this review will be timely and will stimulate discussions in the field about the relevance of such modifications. Although, there exists review papers from the last few years on the topic of ncRNA modifications (Int. J. Mol. Sci. 2021, 22, 6166), this paper will still be relevant if it focuses extensively on how these modifications are related to ncRNA structure, protein binding and R-loop formation. Since these earlier reviews have a much broader focus including the RNA modification enzymes, RNA modification detection methods as well.

Major comments:

There are several major concerns regarding how the manuscript is written, the logic behind the flow of the concepts, the main title of the paper and titles of several sub-sections. In the following paragraph, I have highlighted these concerns.

  • The current title of the paper is misleading with regards to the content of the paper. The current title is very general and suggests that the paper will discuss RNA structure and functions. But the paper goes into detail about how RNA modifications affect their structure and function. This is in fact made evident from sentence 23 of the abstract where the authors state that “…. influenced by epictranscriptome marks is discussed.” The authors can modify the title to something where the term RNA epitranscriptomics or RNA modifications are mentioned.

  • An overall comment on how the introduction section could be re-organized for coherent flow of ideas:

The current format of the introduction section first describes what are epitranscriptomic marks, then introduces lncRNAs, and then talks about how studying lncRNA-binding partners is important. The section ends by stating how these epitranscriptomic marks influences lncRNA structure and function is the aim of this review. An alternative suggestion would be:

  • Introduce what are lncRNAs
  • Describe how they function i.e., interacting with RNA, DNA or proteins.
  • Describe what are RNA modifications and compare mRNA/lncRNA modifications.
  • Then end the section by saying how these epitranscriptomic marks can affect lncRNA function.

  • It has also come to my attention that the abstract written here has been self-copied from the Special Issue Information section of this journal (Refer to Special Issue "Systematic Analysis of lncRNA Structures and Functions"). Kindly consider re-writing the abstract in different words.

  • Kindly rephrase the sentences 55-57 where the authors describe RNA pull downs coupled with LC-MS. Make this statement a bit generic.

One could say “the most popular method is using an affinity tag on an in vitro purified RNA and using this RNA as a bait to pull-down proteins/nucleic acids from cellular extracts.” This is because biotinylated lncRNA are just one examples of the several other in vitro methods such as MS2 tagging, Anti-sense tagging.

  • Since there are a lot of previous literature known about mRNA modifications, it is important to compare the functional significance of mRNA and ncRNA modifications in the introduction section.

  • The title of the second section is mis-leading. The title suggests that this section is about structuredness of RNAs and their role in function. But this section as it is written, focuses more broadly on (i) RNA structure probing methods, (ii) innate immune response pathways, (iii) lncRNAs having RNA modifications, involved in innate immune response and (iv) excessive details on A to I editing. It isn’t clear whether the section #2 is about RNA structure in innate immune response or RNA modifications.

    Why is there an emphasis only on innate immune response if the title suggest “divergent roles”.

Here are some ideas to improve the coherence of the ideas in this section. The main concept in this section could be: “how are RNA structures related to RNA modification sites?”. In sentence 79-81, the authors suggest comparing RNA structuredness to epitranscriptome marks. This section would be more convincing if the authors could do this comparison from published data if available, and include it in a figure accompanying this section.

  • Consider rephrasing the sentence 68. “Unlike DNA, RNA is mostly single stranded”. This is highly context dependent and a debatable sentence. The reference #25 for this sentence is also not accurate. Furthermore, this reference has not been cited at all except for once. So, I would recommend removing this reference and repharsing this opening sentence.

  • In sentences 73-75, the authors only mention PARS. It is important to mention other highly cited methods such as SHAPE-MaP, DMS-Seq etc.

  • Sentence 79-82 is well-written and is an important concept that this review introduces. Please cite some well-known data base of high throughput epictranscriptome marks. Have such an analysis of comparing structuredness with RNA modifications been done elsewhere, even in a low-throughput manner?

  • Why are the authors suddenly talking about innate immune response? The review doesn’t say that it’s going to focus on this topic. Consider removing texts about innate immunity. It only confuses the reader. One recommendation is to use innate immune response as a biological function to describe these idea that RNA structure and epitranscriptomic marks affects the functioning of the RNA.

  • In sentence 92-93, the authors mention m6A plays a role in innate immunity. Write a sentence or two about how does it play a role?

  • Reference #44 only talks about mRNA modifications. Why is this cited for a statement about lncRNA modification?

  • Sentences 95-101 is too long. Consider splitting it into two for ease of understanding.

  • Sentences 102-105 talks about pseudoU. Has this modification been discovered in many lncRNAs? if yes, provide references and examples? If no, then discuss pseudoU in mRNAs in more detail and compare with how pseudo in lncRNAs might be functionally relevant?

  • Why have the authors described in such details about A to I editing enzymes and not m6A enzymes or pseudo enzymes? It confuses the reader since the aim of this review is not about the enzymes that adds these modifications but rather about how these modifications influence lncRNA structure and function.

  • Sections 3 and 4 are well written, and the message is clear. In sentence 133-135, instead of giving an example of a promiscuous binder, consider giving an example of a specific binder to illustrate the importance of protein binding in lncRNA function. One such example would be NORAD and pumilio proteins.

  • In sentences 150-155, the authors use an example of a m5C modification in the 3’-UTR of an mRNA to illustrate the importance of RNA modifications in protein-RNA binding. I recommend using a more direct and well-studied example of HOTAIR 783 m6A modification and how this m6A modification in a lncRNA affects the protein YTHDC1 binding and therefore influence the RNA function.

  • Why is there a sudden mention of R-loops? Could the authors explain why they chose to focus on R-loops especially in a review discussing about how RNA modifications affect RNA structure and function? If so, please provide evidence to suggest lncRNA modifications are involved in R-loops. The references #92-#94 only suggests m6A in mRNAs to be associated with R-loops. If so, then state clearly that the connection between RNA modifications and R-loops are only from studies on mRNAs so far. The authors can then extend this to lncRNAs since they are also transcribed by pol II .,

  • “R-loops as mediators of chromatin …” as the title of section 4 does not flow well with the narrative of this review. The review is about how RNA modifications influences lncRNA structure and function including the ability of lncRNA to form R-loops. So, consider rephrasing the title into something like “Factors influencing lncRNA mediated R-loop formation – sequence, structure and chemical marks”.

  • In sentence 171, cite a reference for pathological effects of improper R-loop regulation.

Minor comments:

  • “…whose number well surpass that of protein-coding genes”. This line in sentence 49 has been repeated before in abstract sentence 15-16. Either remove this in sentence 49 or consider re-wording it.

  • In sentence 96, consider rephrasing “…that might affect their secondary structure as in the case of…”.

  • Sentences 143-149 is a long sentence. It may confuse the readers. Split the sentences into two or three for ease of understanding.

  • The sentence 179-180 is very similar to sentence 175. Consider rephrasing or removing one of these.

  • In sentence 215, explain what is RAD51?

  • In sentence 217, remove ‘lncRNA’ after GAL7.

  • In sentence 222, make in trans into italics.

  • Why haven’t the authors used any cartoons or figures to explain the main messages conveyed in the individual sub-sections? Perhaps using Figures to describe the messages to sub section #2 could clarify what exactly the authors are trying to convey.

  • A general remark – In the relevant sections, include more references for lncRNA modifications and their role in lncRNA biology.
